health and disease and epidemiology/
mathematical modelling

COVID-19, vaccination, switching,
long-term, burden

**Authors for correspondence:**
Ruiyun Li
e-mail: ruiyun.li@ibv.uio.no
Nils Chr. Stenseth
e-mail: n.c.stenseth@mn.uio.no

# Switching vaccination among target groups to achieve improved long-lasting benefits

Ruiyun Li[1], Ottar N. Bjørnstad[1,2] and Nils Chr. Stenseth[1]

[1]Centre for Ecological and Evolutionary Synthesis (CEES), Department of Biosciences, University of Oslo, 0316 Oslo, Norway
[2]Department of Biology, Center for Infectious Disease Dynamics, The Pennsylvania State University, University Park, PA, USA

RL, 0000-0001-8927-9965; ONB, 0000-0002-1158-3753; NCS, 0000-0002-1591-5399

The development of vaccines has opened a way to lower the public health and societal burden of COVID-19 pandemic. To achieve sustainable gains in the long term, switching the vaccination from one target group to a more diverse portfolio should be planned appropriately. We lay out a general mathematical framework for comparing alternative vaccination roll-out strategies for the year to come: single focus groups: (i-a) the high-risk older age groups and (i-b) the core-sociable groups; and two focus groups: (ii-a) mixed vaccination of both the high-risk and core-sociable groups simultaneously and (ii-b) cyclic vaccination switching between groups. Featuring analyses of all relevant data including age pyramids for 15 representative countries with diverse social mixing patterns shows that mixed strategies that result in both direct and indirect protection of high-risk groups may be better for the overall societal health impact of COVID-19 vaccine roll-out. Of note, over time switching the priority from high-risk older age groups to core-sociable groups responsible for heightened circulation and thus indirect risk may be increasingly advantageous.

## 1. Introduction

The SARS-CoV2/COVID-19 pandemic has overwhelmed public health systems and provoked considerable tolls on global economics and societies. The major objectives of management and mitigation of the pandemic are not only to reduce the burden of mortality in the initial waves of the pandemic but also during the likely intermediate term transition towards endemicity [1,2]. One of the many pressing questions is to whom the highest priority

### PUBLISHING

should be given during roll-out in order to maximize the societal benefit of the COVID-19 vaccines. A critical remaining question is when and whether to switch the prioritization among population age/social groups in countries with different age pyramids and age-structured mixing patterns.

The overarching goal of vaccination programmes broadly adopted across the world is to reduce COVID-19-related mortality in the population. Most current policies towards attaining this emphasize vaccinating the groups according to their relative risk of severe disease [3]. This strategy has a proven effectiveness in reducing the burden of mortality in the early stage of vaccine roll-out, in scenarios of low vaccine efficacy or for the essential health workers [4–9]. Yet, historical evidence from other infectious disease challenges suggest that such critical direct protective measures may lose primacy to the benefit of indirect protection in the intermediate and long term [10]. In this time of a pandemic moving towards endemicity, we need mathematical tools for flexible prioritization of vaccine roll-out. Here we study the overall societal health benefits of four allocation strategies, with a particular focus on contrasting strategies with single focus groups: (i-a) the high-risk older age groups and (i-b) the core-sociable group of young amplifiers of circulation, versus strategies with mix prioritization among these two groups: (ii-a) simultaneously (hereafter 'mixed vaccination') and (ii-b) cyclically switching vaccination among the high-risk to the core-sociable ones (hereafter 'cyclic vaccination'). One motivation for a cyclic regime is to keep high-risk and high-contact from visiting vaccine-providing centres at the same time.

# 2. Material and methods

## 2.1. Demography and social mixing patterns

To study the relative effectiveness of vaccine roll-out across the globe, we selected 15 countries covering a broad range of demographic and social mixing patterns, including Australia, China, Finland, France, Germany, Italy, Japan, New Zealand, Norway, South Africa, South Korea, Spain, Sweden, UK and USA (in alphabetical order). For these countries, we collected age pyramids from the statistics of the United Nations [11] and country/age-specific number of contacts from Prem *et al.* [12] and Fumanelli *et al.* [13]. For our analysis, we aggregated the data for each country to 17 5-year age groups (i.e. 0–4, 5–9, …, 75–79, greater than 80 years old); though the fully reproducible model framework presented in detail in [2] is completely flexible to age groupings. The social mixing data allow us to calculate marginal number of contacts for each age group and identify the age brackets with highest socially mixing rates (see below).

## 2.2. Vaccination strategies

The targeted use of vaccine doses is critical to maximize its public health impact. Here, we define four strategies with different deployment of vaccination (electronic supplementary material, figure S1). We initially define two vaccination prioritization strategies, focusing only on either (i) the high-risk older age groups (i.e. groups at high risk of dying) or (ii) the core-sociable ones (i.e. groups at high risk of exposure and onwards transmission). The high-risk older age group includes the elderly over 65 years old, comprising four age groups, i.e. people over 80, 75–79, 70–74 and 65–69 yr. This group, depending on the total vaccine doses, may also include the 45–64 yr groups, and eventually the 25–44 yr groups. Thus we study the currently common strategy of prioritizing the elderly and move to other groups by moving down the age ladder. The core-sociable groups (i-b) are identified according to the marginal distribution of contacts in each age group for each country. We take this to be the four age groups with the highest number of contacts, and possibly the next four age groups with the second highest contacts in scenarios of increased vaccine doses. We further consider vaccinating both the older and sociable groups by allocating half of the total dose to each, and further define two alternative strategies that differ in the timing of vaccination for each group: (ii-a) mixed vaccination which focuses on two groups simultaneously, and the (ii-b) cyclic vaccination that targets two groups sequentially, each of which lasts for half the duration of the campaign.

Together with the various strategies, we consider three frequencies of vaccination roll-out that initialized once only, annually and every six months. For each vaccination campaign, we assume a duration of three months. The one-off vaccination is defined as the single campaign in the initial year without any follow-ups. The annual vaccination is defined as the three-month campaign in the initial year and wait for nine months to start the second campaign. Similarly, for the six-month approach,

we initialize the three-month campaign, followed by a three-month intermittent period and then another vaccination campaign. While we realize these are crude caricatures, we think of them as useful heuristics to highlight the overall conceptual questions of who and when to target different age groups. Of note, we do not distinguish who is still protected by the vaccine and who is not in the following campaigns. We do this deliberately because coronavirus immunity may not be very long-lived, so repeat vaccination may be important. This means that people can get vaccinated irrespective of their vaccine-card status. To lay out the general framework and findings, we assume all-or-nothing vaccine efficacy for which the vaccine provides perfect—though transient—protection to a fraction of individuals (see below). Additionally, novel evidence from mass vaccination campaigns have shown that vaccines could be highly effective against SARS-CoV-2 infections across diverse populations in the real-world setting [14–17]. We therefore assume a strong protection against infections and do not explicitly distinguish the efficacy against infections and that against the progression to severe disease and death. For similar heuristic reasons, we assume that vaccine-induced immunity is as high as that from prior infections and thus the rate of loss of immunity for recovered and vaccinated individuals to be the same.

## 2.3. Model simulations

Building on [2], we developed a realistic age-structured multi-compartmental SEIR model that allows for projections of disease burden of SARS-CoV-2 virus with diverse intervention strategies:

$$\frac{dS_i}{dt} = \omega R_i - \underbrace{\lambda_i S_i}_{\text{infection}} - \underbrace{q Q_i S_i}_{\text{vaccination}} \tag{2.1}$$

$$\frac{dE_i}{dt} = \lambda_i S_i - \underbrace{\delta E_i}_{\text{exposure}} \tag{2.2}$$

$$\frac{dI_i}{dt} = \delta E_i - \underbrace{\gamma I_i}_{\text{recovery}} \tag{2.3}$$

$$\frac{dR_i}{dt} = \gamma I_i + q Q_i S_i - \underbrace{\omega R_i}_{\text{lost immunity}} \tag{2.4}$$

where $S_i$, $E_i$, $I_i$ and $R_i$ are, respectively, the number of susceptible, exposed, infected and recovered individuals in age group $i$. The recovered/vaccinated individuals are assumed to lose immunity and return to susceptibility after an average protected period of $1/\omega$ and subsequently be liable to reinfection. The average incubation period $(1/\delta)$ and infectious period $(1/\gamma)$ in the analysis is taken to be 5.2 and 7 days, respectively [18–20]. The force of infection on susceptibles in age-class $i$ is $\lambda_i = \beta \sum_j^n C_{ij} I_i / N_i$, where $\beta$ is the baseline rate of transmission given by $\beta = R_0 \gamma$ and $C_{ij}$ is the normalized contact rate between age group $i$ and $j$ [2]. To appropriately model the *fraction* of individuals vaccinated, we define the *rate* of vaccination $(Q_i)$ as $-\log(1 - P_i)/D_i$ where $P_i$ and $D_i$ are the vaccine coverage and duration of vaccination in age group $i$, respectively [21]; $q$ is the vaccine efficacy.

Given the realistic age-structured SEIR equations defined by equations ((2.1)–(2.4)), we numerically integrate the model to predict dynamics of COVID-19 over the next 20 years using a variety of scenarios. To illustrate the conceptual questions we wish to address, we assume that the total vaccine dose can cover 50% of each country-specific population. This overall coverage is then translated to the group-specific coverage based on strategy (i-a) – (ii-b). Assuming a 1-year average duration of immunity, we initially simulated the model with UK parameters as an example, in plausible scenarios with varying levels of transmission ($R_0 = 1.5$, 2 and 3) and efficacy of vaccine ($q = 20\%$, 50% and 80%); While 20% efficacy is not likely to be relevant for the COVID-19 situation, it extends our considerations to be relevant for, for example, influenza vaccination in years of mismatch. For each scenario, we project trajectories under various vaccination strategies and frequencies. To examine the effect of immunity durations, we further simulate the model with alternative assumptions of immunity lasting six month, 5 years and permanently; again the assumption of permanent immunity is unlikely for the novel coronavirus [1], but it serves to illustrate how future long-term sterilizing vaccine technology may affect disease dynamics. We apply the model to 15 countries by parametrizing with various demographies and social mixing patterns to reveal the effect of country-specific context. Across countries, simulations are initialized with 99.9% and 0.1% of the susceptible and infected individuals in the population, respectively. This helps clarify how epidemic trajectories vary with country-specific demography, mixing and vaccine roll-out.

We estimate the population-level fraction of infections and mortality in the short term (i.e. at the end of the first six months, and the first and second year) and long term (i.e. at the end of the 5th, 10th and 20th year) for each scenario and country. We calculate mortality by multiplying the modelled infected fractions and the current reports on the age-specific infection–fatality ratio [22], recognizing that prior exposure may and/or may not modulate such patterns following transitions to endemicity [1,2,23]. We further calculate the projected reduction of mortality and infections as compared with the baseline in the absence of vaccination. With these estimates, we first compare the direct and indirect benefits from the direct vaccination of the high-risk older age groups or high-contact groups versus the mixed vaccination of both the high-risk and core-sociable groups. Moving beyond the mixed vaccination, we further consider alternative approach to leverage both the direct and indirect protection. To do this, we examine the effectiveness of simultaneously or cyclically vaccinating the groups. We compare projected gains by using cyclic strategy and strategies with vaccination prioritized to the elderly only and the core sociable only in the short and long term. We provide sensitivity insights to our conclusions by examining vaccine supplies covering 40% and 90% of the populations, and the impact of the changing social behaviour during the pandemic, such as the lower contact rate of the elderly, by using the age-specific contact matrix for 2020 [24].

# 3. Results

## 3.1. Benefits from direct and indirect protection

We first examine effectiveness of strategies in a setting with the same demography and social mixing patterns as reported for the UK, assuming a total vaccine allotment capable of covering 50% of the population (for other countries and doses, see electronic supplementary material). Across various immunity durations we demonstrate that the mixed vaccination, allocating half of the dose to the high-risk and the other half to the core-sociable ones, would lead to a considerably greater reduction of the burden of mortality than vaccinating the older age groups and focusing on the direct risk only (figure 1; electronic supplementary material, figure S2). However, this incremental gain from indirect protection is predicted to vary across levels of transmissibility, vaccine efficacy and frequency of (re)vaccination (table 1 and electronic supplementary material, tables S1–S3). Assuming a 1-year immunity duration, we estimate that indirect protection would contribute to approximately 43% and 35% incremental reduction of mortality in low-level transmission settings ($R_0 = 1.5$) at the end of the first six months by using a low-to-medium (20%, 50%) and high (80%) efficacious vaccine, respectively. The gains decrease with the level of transmission and over time. This result is suggestive of the need for frequent revaccination in settings with medium-to-high transmissions in subsequent years. With baseline parameters, we predict an additional approximately 12% reduction of mortality in low-transmission settings at the end of the second year when initializing an annual vaccine campaign as compared to one-off vaccination. An annual vaccination initiative, however, would only lead to marginal increase in the reduction of mortality burden, particularly by the end of the second year in medium-to-high transmission ($R_0 = 2$ and 3) settings. Scaling up vaccination frequency to every six months would (given baseline parameters) give rise to a median of 7.4% (95% CI: 2.6–11.3%) and 11.9% (9.4–19.5%) less mortality than the one-off initiative at the end of the first and second year, respectively. Of note, our projections for a range of countries across the globe suggest that regardless of demography and mixing, prioritizing vaccination to groups at health risk while accounting for the indirect protection of those groups by vaccinating high transmission groups is critical (electronic supplementary material, figure S3). Additionally, the benefits attributable to indirect protection are predicted to be broadly consistent across various assumptions of the total doses available (electronic supplementary material, figure S4). This finding also applies to the pandemic with the changing social behaviours and, in particular, the lowered contacts with the elderly (electronic supplementary material, figure S5).

## 3.2. Effectiveness of cyclic vaccination

Temporal separation of vaccination of the two groupings may be important to minimize exposure of high-risk groups from high-contact groups. To highlight this conceptual issue, and how our broad mathematical framework can address it, we lay out a caricature of the notion of cyclic vaccination during the transition from the virgin pandemic to endemicity [1,2]. We first estimate the reduction of

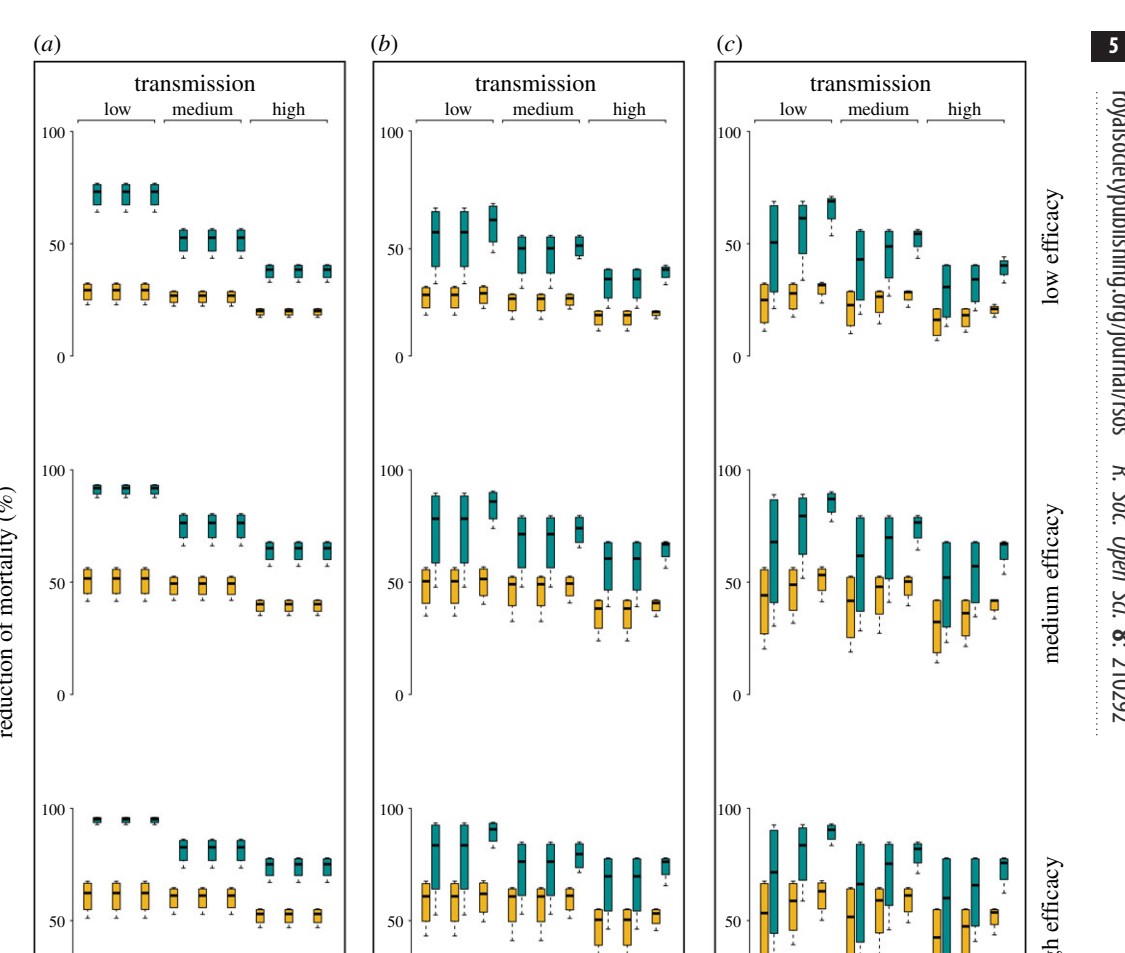

**Figure 1.** Effectiveness of direct and indirect protection in reducing the burden of mortality. These simulations use demographic and social mixing patterns for the UK, together with varying immunity durations (six-month, one-year, five-year, permanent), levels of transmission ($R_0 = 1.5$, 2 and 3), vaccine efficacies (20%, 50% and 80%) and vaccination frequencies (one-off, annual, every six months). Median estimates and 95% CI across immunity durations are shown for scenarios with differing transmission level, vaccine efficacy and vaccination frequency. Direct protection is simulated by (i-a) vaccinating the high-risk older age groups only; while the indirect protection is simulated by (ii-a) the mixed strategy allocating half dose to the high-risk and the other half to the core-sociable groups simultaneously. Reduction of mortality as compared to no vaccination is estimated at the end of the first six months (a), the first (b) and the second (c) year since the initiation of vaccination. For the baseline, the total vaccine supply is assumed capable of covering 50% of the population.

mortality by the cyclic strategy and vaccinating the high-risk older age groups in the short and long term. Overall, our model predictions suggest a substantial reduction in the overall mortality by using cyclic vaccination as compared to focusing on the high-risk groups only (figures 2 and 3). Across efficacies and transmission intensities, allocating half the doses to the older individuals and subsequently the other half to the core-sociable ones is predicted to lead to a median reduction of 26.4% (18.3–35.9%) less mortality at the end of the first six months (electronic supplementary material, table S4). Over time the effectiveness of cyclic vaccination is strongly dependent on the frequency of vaccination campaigns. If initiated once, the short- and long-term effectiveness of the cyclic vaccination would be considerably lower, but nevertheless reducing the burden of mortality by a median of 14.3% (9–15.7%) and 1.8% (1.1–2.2%) by the end of the 2nd and 20th year across efficacies and transmissions. Initializing the cyclic vaccination annually would have similar contribution in medium-to-high transmission settings, with a median of 18.2% (13.8–19.6%) and 15.1% (4.7–21.7%) less mortality by

**Table 1.** Incremental reduction of mortality from indirect protection. These simulations are the same as in figure 1, but summarize the incremental reductions of mortality that are attributable to indirect protection as compared to direct protection, assuming a 1-year immunity duration. Estimates under alternative immunity duration are presented in the electronic supplementary material.

| year | vaccine efficacy | one-off vaccination | | | annual vaccination | | | vaccination every six months | | |
|---|---|---|---|---|---|---|---|---|---|---|
| | | level of transmission | | | level of transmission | | | level of transmission | | |
| | | low | medium | high | low | medium | high | low | medium | high |
| 0.5 | low | 43.37 | 24.46 | 17.69 | 43.37 | 24.46 | 17.69 | 43.37 | 24.46 | 17.69 |
| | medium | 42.42 | 26.36 | 24.20 | 42.42 | 26.36 | 24.20 | 42.42 | 26.36 | 24.20 |
| | high | 35.78 | 21.41 | 21.72 | 35.78 | 21.41 | 21.72 | 35.77 | 21.41 | 21.72 |
| 1 | low | 23.90 | 20.58 | 14.46 | 23.89 | 20.58 | 14.45 | 31.30 | 22.67 | 21.92 |
| | medium | 23.20 | 18.42 | 18.85 | 23.20 | 18.42 | 18.85 | 34.67 | 22.91 | 26.58 |
| | high | 19.31 | 11.50 | 16.11 | 19.31 | 11.50 | 16.11 | 30.39 | 17.19 | 23.53 |
| 2 | low | 17.35 | 14.26 | 10.08 | 32.85 | 18.34 | 12.69 | 37.61 | 25.38 | 21.89 |
| | medium | 17.67 | 13.91 | 13.90 | 29.93 | 17.64 | 16.25 | 33.99 | 26.16 | 24.73 |
| | high | 14.06 | 9.71 | 12.52 | 25.15 | 13.14 | 14.08 | 28.69 | 21.65 | 21.56 |

the end of the 2nd and 20th year, respectively. Of note, the model predicts that if implemented every six months, the cyclic vaccination is typically effective over the course of long-term disease dynamics. We estimate a median of 30.5% (25.5–36.4%) and 22.3% (15.5–29.8%) of gains achievable in low and medium-to-high transmission settings, respectively. These results are suggestive of a sustained effectiveness by initializing the cyclic vaccination strategy in a frequent manner.

We further predict the short- and long-term effectiveness of cyclic vaccination in reducing the burden of mortality relative to targeting groups at high risk of transmission. Across model scenarios, we see that switching the vaccination would contribute to substantial reduction of mortality as compared to focusing on the core-sociable only (electronic supplementary material, table S5). By the end of the 2nd year, we estimate a median incremental reduction of mortality of 27.1% (23.0–39.0%), 36.8% (27.1–51.3%) and 41.3% (27.7–57.7%) by switching the vaccination once, annually or every half year, respectively. Additionally, the long-term incremental effectiveness is highly dependent on the frequency of vaccination initiatives. We predict that the one-off cyclic vaccination would lead to a less substantial lowering of the burden of mortality by a median of 4.3% (3.4–5.6%) at the end of the 20th year. By contrast, accelerating the cyclic vaccination initiative in the long term would sustain the short-term gains, with a median of 34.7% (25.9–46.4%) and 42.7% (32.8–56.8%) greater reduction of mortality if implemented once and twice a year, respectively.

# 4. Discussion and conclusion

The overall reduction in public health and societal burden is an optimization problem to balance direct and indirect protection through vaccinating the high-risk and high-contact ones. The targeted allocation of limited vaccines to prioritized populations is therefore a central pillar of maximizing the public health impact of limited supplies. Of note, using a variety of vaccine designs has been adopted for the control of other viruses. For example, for measles combined strategies of routine vaccination, intermittent supplementary immunization activities (SIAs) and outbreak response vaccination has been repeatedly deployed in measles endemic areas [25]. SIAs are also used to supplement routine vaccination for the polio end-game [26]. By evaluating effectiveness of various strategies, we point to the considerable gains in lowering the burden of mortality that are attributable to indirect protection. Adding values to the indirect protection strategies are their greater contribution to reducing the overall number of infections as compared with those from direct protection of high-risk but often low-contact groups (electronic supplementary material, figure S6). A previous study has shown that direct protection through vaccinating the elderly would moderate the burden of mortality but have marginal contribution to lowering the total number of infections [6]. Consistent with this, we emphasize that

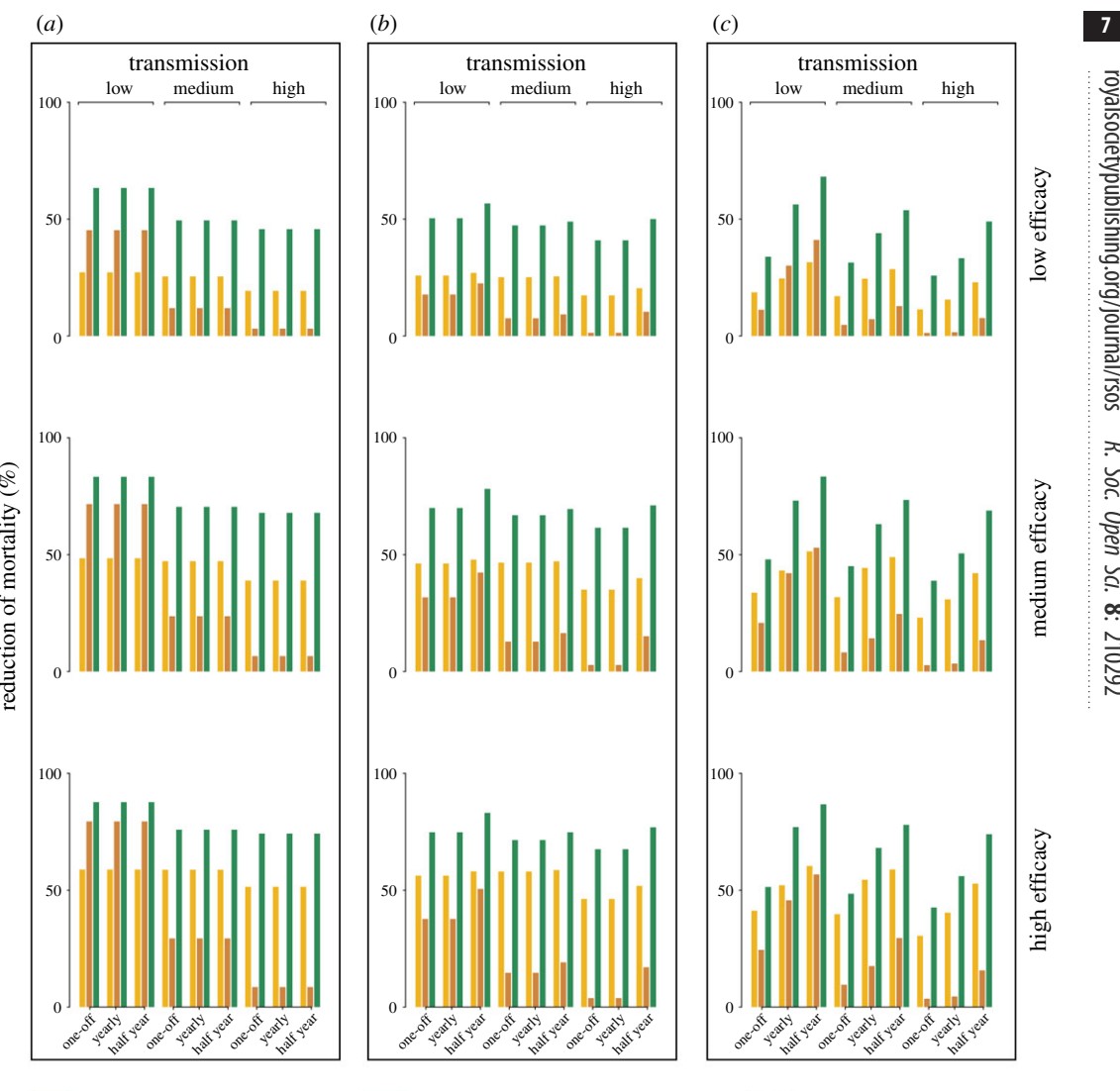

**Figure 2.** Relative effectiveness of different vaccination strategies in the short term. These simulations use demographic and social mixing patterns for the UK, together with the 1-year immunity duration and varying levels of transmission ($R_0 = 1.5$, 2 and 3), vaccine efficacies (20%, 50% and 80%) and vaccination frequencies (one-off, annual, every six months). We consider three strategies: (i-a) vaccinating the high-risk older age groups only, (i-b) vaccinating the core-sociable groups only, and (ii-b) switching the vaccination from the high-risk to the core-sociable groups. For each strategy, short-term effectiveness is evaluated by the reduction of mortality as compared with no vaccination at the end of the first six months (a), the first (b) and the second (c) year since the initialization of vaccination.

focusing on the high-risk and the core-sociable ones simultaneously would make better use of the joint benefits from both direct and indirect protection, achieving considerable overall public health gains.

By carefully tailoring our model to the COVID-19 context, our findings are relatively specific to COVID-19. Most importantly, our analyses highlight that shifting vaccination priority from the high-risk older ones to the core-sociable groups could have profound and long-lasting effect in reducing disease burden. Over the course of disease invasion, there could be a shift of COVID-19 risks to younger age-classes in future endemic circulation [2]. In the light of this, our model predictions suggest that switching the vaccination prioritization in a highly frequent manner could be very beneficial. In particular, if assuming a 1-year duration of immunity, we suggest that carrying out the cyclic vaccination every six months since the second year of initializing the vaccination campaign would be a more ideal approach to achieve approximately 21% (low-level transmission) and 9% (medium-to-high transmission) lower burden of mortality when compared with targeting the older ones only in the long term. While vaccine response could be all-or-nothing or leaky, we believe

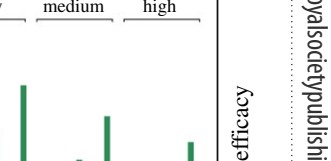

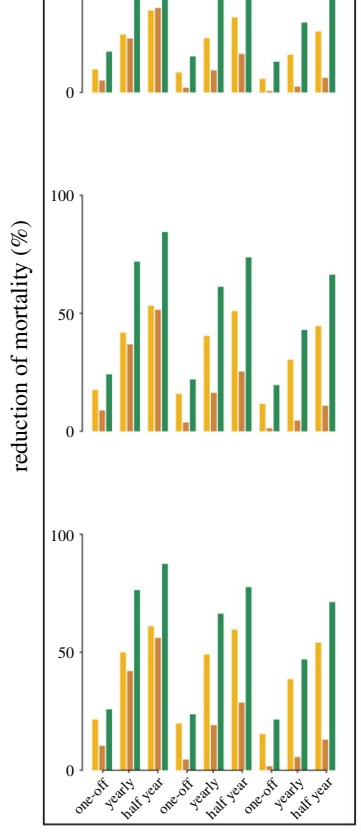
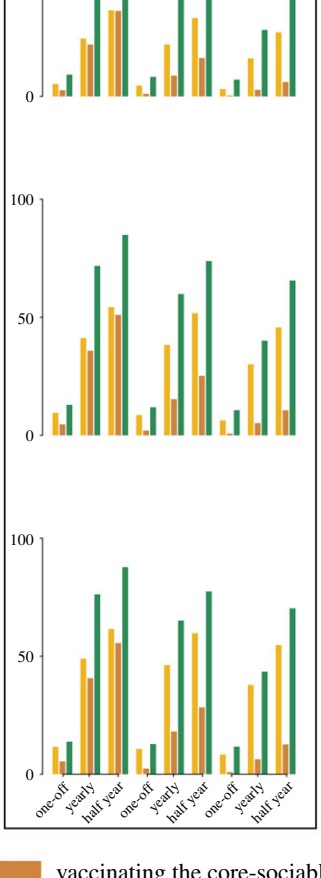
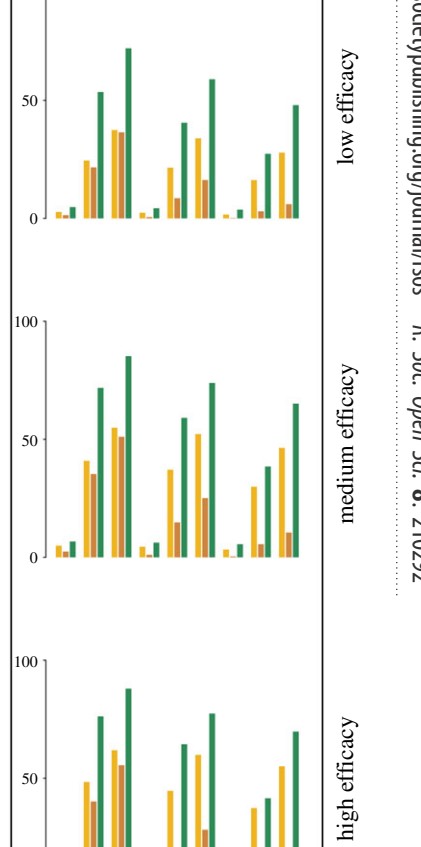

vaccinating the high-risk    vaccinating the core-sociable    cyclic vaccination

**Figure 3.** Relative effectiveness of different vaccination strategies in the long term. These simulations are the same as in figure 2, but considering the long-term effectiveness as quantified at the end of the 5th (*a*), the 10th (*b*) and the 20th (*c*) year.

that the various nuances of vaccine-induced immunity [27] will not modulate our findings and implications, though the explicit response of the new mRNA adenovectored vaccine technology is a crucial issue for future study.

Our study has two main goals. First is to offer insights that can help guide COVID-19 vaccination roll-out in both high- and low-income and high- and low-transmission settings beyond the initial pandemic and across the world to iteratively advance the vaccination campaigns in the years to come. The second goal is to introduce an open-source computational framework for scenario analyses of optimal vaccine roll-out for age-structured communities in the face of future public health challenges.

Data accessibility. The data and code that support the findings in this study have been made openly available at https://github.com/ruiyunli90/Corona-vaccine.

Authors' contributions. R.L., O.N.B. and N.C.S. conceptualized the study. R.L. investigated the models, made the analyses and led the writing. R.L., O.N.B. and N.C.S. interpreted the results and developed the manuscript.

Competing interests. We declare we have no competing interests.

Funding. This work is funded by the Research Council of Norway COVID-19 Seasonality Project 312740 (N.C.S.) and the Penn State University Seed-Funded COVID-19 Project (O.N.B.). We thank two anonymous reviewers for the insightful comments and suggestions.

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
