## [Peer Review File · Royal Society Open Science]

Review History

RSOS-210292.R0 (Original submission)

Review form: Reviewer 1

Is the manuscript scientifically sound in its present form?

Yes

Are the interpretations and conclusions justified by the results?

No

Is the language acceptable?

Yes

Do you have any ethical concerns with this paper?

No

Have you any concerns about statistical analyses in this paper?

No

Recommendation?

Major revision is needed (please make suggestions in comments)

Comments to the Author(s)

This paper compares some vaccination strategies with the intent of reducing the mortality of COVID-19.

The approach seems reasonable and the paper is clearly written and well structured.

My main concerns regard the assumptions made by the authors, since I believe the results strongly depend on them and I am not fully convinced that this setup is the most significant one:

- why is the low efficacy vaccine only 20% effective? It seems that vaccines having less than 60% efficacy will not be approved, at least in Europe. Moreover, most approved vaccines have an efficacy to prevent severe disease which is 100% or very close to that.

- I think that for this study it is fundamental to distinguish how much a vaccine protects against the severe disease and how much a vaccine protects against becoming infectious. Since it seems that the two numbers are different, this distinction cannot be avoided. Moreover, many currently available vaccines seem to have 100% efficacy (or close to that) against severe disease, while lower (even much lower, around 60-70% for some) efficacy against light disease, which might entail that the infected person is infectious. I think it is still a subject of debate to what extent currently available vaccines protect also against transmitting the infection. If this information were included in the proposed analysis (by taking some reasonable assumptions, given the lack of results), I am confident that the standard strategy of aiming at direct protection will turn out to be the best.

- I am not sure why the authors assume that only 50% of the population will be vaccinated, since most countries are aiming at vaccinating 100% of the population. Also, it is not fully clear to me what the different vaccination policies are. Maybe you can include an equation or an algorithm to explain that. Does one-off mean that you vaccinate 50% of the population and then stop vaccinating forever? For the yearly campaign does that mean that in 3 months you vaccinate 50% of the population, then wait for 9 months and start vaccinating again? In that case, do you assume to know who is still protected by the vaccine and who is not? It seems hard to make this distinction with the model you use, since some (few) people lose protection almost immediately with model (1)-(4). The same question is even stronger for the 6-month approach, since in that case you should have a moment in which 100% of the population is protected. Then how can the three strategies be distinguished? I think this point needs to be clearly explained since it is crucial to understand the results.

Minor comments:

- In (4) I suppose a "q" is missing, otherwise the total population is not preserved constant.

- planed -> planned

- I disagree with the following conclusion: "In particular, if assuming a 1-year duration of immunity, we suggest that carrying out the cyclic vaccination every six month since the second year of initializing the vaccination campaign would be optimal to achieve approximately 21% (low-level transmission) and 9% (medium-to-high transmission) lower burden of mortality as compared to targeting the elderly only in the long-term."

Left aside the discussion on the validity of the assumptions, the only claim that can be made is that one of the studied strategies is better than the other studied ones. In order to claim

optimality, one needs to formulate and solve an optimization problem, or to prove that (at least locally), there exists no strategy which performs better.

Review form: Reviewer 2

Is the manuscript scientifically sound in its present form?

No

Are the interpretations and conclusions justified by the results?

No

Is the language acceptable?

Yes

Do you have any ethical concerns with this paper?

No

Have you any concerns about statistical analyses in this paper?

No

Recommendation?

Major revision is needed (please make suggestions in comments)

Comments to the Author(s)

This work compared different age-based vaccination strategies for vaccines in different countries, for endemicity control. The paper is clearly written and to the point. The code is clear as well, and the findings are reproducible.

I do have major concerns that advise for at least major revision.

- The model assumes an equal absolute amount of contact for each age-class, which in this case influence the results greatly. The death process is not defined in the main text.
- Some assumptions, mainly regarding vaccine roll out function, initial conditions, are counterfactual. And the conclusion is dependent on these. However, the writing overstates the generalization and the practical importance of the findings. The scenarios seem to spend a different total amount of vaccine per day, making the comparison result obvious: the more we vaccinate the more it is effective (I'm not sure of that, so maybe the authors might correct me on this).

Abstract:

Except if derived from the journal editorial policy, I think it would be interesting to have a hint into the results. The last sentence of the abstract may be re-written to be more comprehensible.

Introduction

L39: successive waves, as some of the 15 studied countries had more waves.

L46: this can be argued to be DALYs too (e.g. avoid two 95 y.o. deaths may not be considered more important than avoiding a 40 y.o. death). Doesn't matter in this context though.

L50: Matraj et al (somewhere on medRxiv last time I looked) also tackle this issue and are not cited in the paper (I'm not affiliated to them in any way).

The introduction is short and to the point, which is great. I do believe a bit more of literature could be cited on this kind of problem (vaccination of high transmission vs high death).

Methods:

The two papers cited for age-specific contact are state of the art, however there has been some changes with the pandemic awareness, lowering the contact of the elder. It would be interesting (though not necessary for me) to compare to these. One source is <https://cmmid.github.io/topics/covid19/synthetic-contact-matrices.html> (again, not affiliated to them in any way).

L19: this wide prioritization for countries specified, e.g France focused more on the 75+ initially. This undermines a bit the i strategy, because authors consider

L25: this sentence was not immediately clear to me. In fact, I think this whole paragraph should be made clear. Assumptions on vaccine efficacy and duration of protection may be stated here. You should also state here than you consider an all or nothing vaccine (by opposition of a leaky vaccine). This is not necessarily the consensus for COVID vaccines.

The epidemic model is sound and does not consider useless features for the research questions, which is really good. The R_0 seems plausible for endemicity.

I may be wrong, but it seems that authors normalize the contact matrix, and from L46 it seems that beta isn't age-dependent. Hence, an assumption is made that every age-class has the same absolute number of contact (and with whom these contacts are embedded in matrix C). This assumption is not realistic and some class have order of magnitude the amount of contact (elderly being notoriously and unfortunately very isolated in these first-world countries, it is visible in these POLYmod matrices).

From the code, I gather that the model is initialized with .1% infected and zero recovered. This is quite unrealistic and should be at least stated in the main text.

Moreover, the death process and its dependence on age should be described.

Results

Depends on the assumptions: when there is no more susceptible because everyone in the age-class is vaccinated, vaccines are wasted in this model and this is not likely to happen in reality. Countries keep a basic accounting of attribution.

The cyclically switching should be done reactively to the proportion of vaccinated in each category (i.e, to keep the spirit of the paper, after the most lethal age-class is 100% (or 75% or 50%, that could be scenarios) vaccinated, move to the next age-class.

The introduction teases a comparison between cyclical switching and mixed attribution, I have not seen these results.

I might be wrong (in which case this should be addressed in the text) but the main result here is dependent on the total throughput of vaccine, which seems different (from the methods) for the different for the scenarios considered. The practical vaccination problem a country has is: given a amount, where to allocate it. Obviously if the amount was dependent, vaccinating across every category in // is optimal. However, as of today, if a dose is given to an 80 y.o, it won't be given to a 40y.o.

Aside from the comment, it would be interesting to see a timeline of vaccination and epidemic in time, to see how the system is reacting and transitioning, in one setup. Like a graph with death and infection as Y and time as X, with vaccine rollout and % vaccinated.

Hence, I would advise to display clearly, for each scenario and each category, how many doses are spent. If it is not equal, to change the problem so that it is equal across scenarios.

Decision letter (RSOS-210292.R0)

Dear Dr Stenseth

The Editors assigned to your paper RSOS-210292 "Cyclically switching target vaccination groups to achieve long-lasting benefits" have now received comments from reviewers and would like you to revise the paper in accordance with the reviewer comments and any comments from the Editors. Please note this decision does not guarantee eventual acceptance.

Please submit your revised manuscript and required files (see below) no later than 21 days from today's (ie 31-Mar-2021) date. Note: the ScholarOne system will 'lock' if submission of the revision is attempted 21 or more days after the deadline. If you do not think you will be able to meet this deadline please contact the editorial office immediately.

on behalf of Professor Enrico Bertuzzo (Associate Editor) and Pete Smith (Subject Editor)
openscience@royalsociety.org

Associate Editor Comments to Author (Professor Enrico Bertuzzo):

The manuscript has now been reviewed by two experts on the field. Both found that paper clearly written and sound. However, they both raised concerns about some hypotheses and asked to clarify some others. Both encourage a major revision to improve, or else justify, the analysis and I share their view.

Reviewer comments to Author:

Reviewer: 1

Comments to the Author(s)

This paper compares some vaccination strategies with the intent of reducing the mortality of COVID-19.

The approach seems reasonable and the paper is clearly written and well structured.

My main concerns regard the assumptions made by the authors, since I believe the results strongly depend on them and I am not fully convinced that this setup is the most significant one:

- why is the low efficacy vaccine only 20% effective? It seems that vaccines having less than 60% efficacy will not be approved, at least in Europe. Moreover, most approved vaccines have an efficacy to prevent severe disease which is 100% or very close to that.

- I think that for this study it is fundamental to distinguish how much a vaccine protects against the severe disease and how much a vaccine protects against becoming infectious. Since it seems that the two numbers are different, this distinction cannot be avoided. Moreover, many currently available vaccines seem to have 100% efficacy (or close to that) against severe disease, while lower (even much lower, around 60-70% for some) efficacy against light disease, which might entail that the infected person is infectious. I think it is still a subject of debate to what extent currently available vaccines protect also against transmitting the infection. If this information were included in the proposed analysis (by taking some reasonable assumptions, given the lack of results), I am confident that the standard strategy of aiming at direct protection will turn out to be the best.

- I am not sure why the authors assume that only 50% of the population will be vaccinated, since most countries are aiming at vaccinating 100% of the population. Also, it is not fully clear to me what the different vaccination policies are. Maybe you can include an equation or an algorithm to explain that. Does one-off mean that you vaccinate 50% of the population and then stop vaccinating forever? For the yearly campaign does that mean that in 3 months you vaccinate 50% of the population, then wait for 9 months and start vaccinating again? In that case, do you assume to know who is still protected by the vaccine and who is not? It seems hard to make this distinction with the model you use, since some (few) people lose protection almost immediately with model (1)-(4). The same question is even stronger for the 6-month approach, since in that case you should have a moment in which 100% of the population is protected. Then how can the three strategies be distinguished? I think this point needs to be clearly explained since it is crucial to understand the results.

Minor comments:

- In (4) I suppose a "q" is missing, otherwise the total population is not preserved constant.

- planed -> planned

- I disagree with the following conclusion: "In particular, if assuming a 1-year duration of immunity, we suggest that carrying out the cyclic vaccination every six month since the second year of initializing the vaccination campaign would be optimal to achieve approximately 21% (low-level transmission) and 9% (medium-to-high transmission) lower burden of mortality as compared to targeting the elderly only in the long-term."

Left aside the discussion on the validity of the assumptions, the only claim that can be made is that one of the studied strategies is better than the other studied ones. In order to claim optimality, one needs to formulate and solve an optimization problem, or to prove that (at least locally), there exists no strategy which performs better.

Reviewer: 2

Comments to the Author(s)

This work compared different age-based vaccination strategies for vaccines in different countries, for endemicity control. The paper is clearly written and two the point. The code is clear as well, and the findings are reproducible.

I do have major concerns that advise for at least major revision.

- The model assumes an equal absolute amount of contact for each age-class, which in this case influence the results greatly. The death process is not defined in the main text.
- Some assumptions, mainly regarding vaccine roll out function, initial conditions, are counterfactual. And the conclusion is dependent on these. However, the writing overstates the generalization and the practical importance of the findings. The scenarios seems to spends a different total amount of vaccine per day, making the comparison result obvious: the more we vaccinate the more it is effective (I'm not sure of that, so maybe the authors might correct me on this).

Abstract:

Except if derived from the journal editorial policy, I think it would be interesting to have a hint into the results. The last sentence of the abstract may be re-written to be more comprehensible.

Introduction

L39: successive waves, as some of the 15 studied countries had more waves.

L46: this can be argued to be DALYs too (e.g avoid two 95 y.o deaths may not be considered more important than avoiding a 40 y.o death). Doesn't matter in this context though.

L50: Matraj et al (somewhere on medRxiv last time I looked) also tackle this issue and are not cited in the paper (I'm not affiliated to them in any way).

The introduction is short and to the point, which is great. I do believe a bit more of literature could be cited on this kind of problem (vaccination of high transmission vs high death).

Methods:

The two papers cited for age-specific contact are state of the art, however there has been some changes with the pandemic awareness, lowering the contact of the elder. It would be interesting (trough not necessary for me) to compare to these. One source is <https://cmmid.github.io/topics/covid19/synthetic-contact-matrices.html> (again, not affiliated to them in any way).

L19: this wide prioritization for countries specified, e.g France focused more on the 75+ initially. This undermines a bit the i strategy, because authors consider

L25: this sentence was not immediately clear to me. In fact, I think this whole paragraph should be made clear. Assumptions on vaccine efficacy and duration of protection may be stated here. You should also state here than you consider an all or nothing vaccine (by opposition of a leaky vaccine). This is not necessarily the consensus for COVID vaccines.

The epidemic model is sound and does not consider useless features for the research questions, which is really good. The R_0 seems plausible for endemicity.

I may be wrong, but it seems that authors normalize the contact matrix, and from L46 it seems that beta isn't age-dependent. Hence, an assumption is made that every age-class has the same absolute number of contact (and with whom these contacts are is embedded in matrix C). This assumption is not realistic and some class have order of magnitude the amount of contact (elderly being notoriously and unfortunately very isolated in these first-world countries, it is visible in these POLYmod matrices).

From the code, I gather that the model is initialized with .1% infected and zero recovered. This is quite unrealistic and should be at least stated in the main text.

Moreover, the death process and its dependence on age should be described.

Results

Depends on the assumptions: when there is no more susceptible because everyone in the age-class is vaccinated, vaccines are wasted in this model and this is not likely to happen in reality. Countries keep a basic accounting of attribution.

The cyclically switching should be done reactively to the proportion of vaccinated in each category (i.e. to keep the spirit of the paper, after the most lethal age-class is 100% (or 75% or 50%, that could be scenarios) vaccinated, move to the next age-class.

The introduction teases a comparison between cyclical switching and mixed attribution, I have not seen these results.

I might be wrong (in which case this should be addressed in the text) but the main result here is dependent on the total throughput of vaccine, which seems different (from the methods) for the different for the scenarios considered. The practical vaccination problem a country has is: given a amount, where to allocate it. Obviously if the amount was dependent, vaccinating across every category in // is optimal. However, as of today, if a dose is given to an 80 y.o, it won't be given to a 40y.o.

Aside from the comment, it would be interesting to see a timeline of vaccination and epidemic in time, to see how the system is reacting and transitioning, in one setup. Like a graph with death and infection as Y and time as X, with vaccine rollout and % vaccinated.

Hence, I would advise to display clearly, for each scenario and each category, how many doses are spent. If it is not equal, to change the problem so that it is equal across scenarios.

===PREPARING YOUR MANUSCRIPT===

a 'clean' version of the new manuscript that incorporates the changes made, but does not highlight them. This version will be used for typesetting if your manuscript is accepted. Please ensure that any equations included in the paper are editable text and not embedded images.

===PREPARING YOUR REVISION IN SCHOLARONE===

- If you are requesting a discretionary waiver for the article processing charge, the waiver form must be included at this step.
- If you are providing image files for potential cover images, please upload these at this step, and inform the editorial office you have done so. You must hold the copyright to any image provided.
- A copy of your point-by-point response to referees and Editors. This will expedite the preparation of your proof.

- Ensure that your data access statement meets the requirements at <https://royalsociety.org/journals/authors/author-guidelines/#data>. You should ensure that you cite the dataset in your reference list. If you have deposited data etc in the Dryad repository, please include both the 'For publication' link and 'For review' link at this stage.
- If you are requesting an article processing charge waiver, you must select the relevant waiver option (if requesting a discretionary waiver, the form should have been uploaded at Step 3 'File upload' above).
- If you have uploaded ESM files, please ensure you follow the guidance at <https://royalsociety.org/journals/authors/author-guidelines/#supplementary-material> to include a suitable title and informative caption. An example of appropriate titling and captioning may be found at https://figshare.com/articles/Table_S2_from_Is_there_a_trade-off_between_peak_performance_and_performance_breadth_across_temperatures_for_aerobic_scop_e_in_teleost_fishes_/3843624.

Author's Response to Decision Letter for (RSOS-210292.R0)

See Appendix A.

RSOS-210292.R1 (Revision)

Review form: Reviewer 1

Is the manuscript scientifically sound in its present form?

Yes

Are the interpretations and conclusions justified by the results?

No

Is the language acceptable?

Yes

Do you have any ethical concerns with this paper?

No

Have you any concerns about statistical analyses in this paper?

No

Recommendation?

Reject

Comments to the Author(s)

The authors did answer on the technical aspects of my criticism, but they did not justify their assumptions, which I still find not realistic and which entail that the conclusions might be the opposite of the conclusions that one would draw if proper assumptions were made.

"we do not explicitly distinguish the efficacy against infectiousness and progression to severe cases and death": this does not answer my criticism on the fact that it seems more and more likely that most vaccines have a very strong protection against severe symptoms and death, while a rather low protection against being infectious. This is one key aspect of the analysis of this paper and cannot be neglected. I think that one cannot just assume that these two numbers coincide and I would rather expect a study on how several ratios impact the results obtained in this paper.

"To provide a general model and findings, we assume an all-or-nothing vaccine which provides perfect protection to a fraction of the vaccinated individuals.": I do not understand how making a (strong and most likely wrong) assumption would make the results more general. To me this is an arbitrary choice which limits the validity of your results to the case in which that assumption is correct. Since all evidence is telling that this assumption is likely to be wrong, I have to conclude that the obtained results do not apply to the COVID-19 pandemic.

"Additionally, to what extent vaccines could reduce the burden of mortality is also dependent on the vaccines supply. Even though prioritizing the at-risk groups is broadly adopted across countries, vaccine supply, in reality, varies vastly across countries. As expected, countries with sufficient doses to cover at least the at-risk groups would lead to a low level of mortality. In contrast, it is less likely to lower the burden of mortality in countries with very limited doses. Acknowledging this uneven distribution of vaccines, we keep a general structure of our model framework without explicitly distinguishing the efficacy against infectiousness and progression to severe cases and death.": I do not see any connection between this reasoning, which I share, and my criticism. If the vaccine supply is sufficient, then vaccinating people at risk is reducing mortality. If the vaccine supply is insufficient then the mortality will be higher. How does this relate to the findings of the manuscript?

Review form: Reviewer 2**Is the manuscript scientifically sound in its present form?**

Yes

Are the interpretations and conclusions justified by the results?

Yes

Is the language acceptable?

Yes

Do you have any ethical concerns with this paper?

No

Have you any concerns about statistical analyses in this paper?

No

Recommendation?

Major revision is needed (please make suggestions in comments)

Comments to the Author(s)

I want to thank the authors for taking the time to account our reviews. I believe my points have been addressed, the (most) major one was that in the previous scenarios' definition, the total number of vaccines given was different between scenario (waste of doses). That's fixed now. Thank you.

Thanks for adding figure S1, which help a lot understanding the scenarios. Thanks also for adding figure S2. I believe Figure S1 & S2 are really important in judging why the scenario's impacts are so different. In S2, the two waves pattern exhibited due the initial conditions, with a really big first wave make the timing of the vaccination of high transmission very important, and drives the obtained results, short and long term. In Figure S1, Scenario A and B are suboptimal because they waste doses in on individuals with low transmission and fatality rate (40 to 65, as fatality comes from Verity et al) in their second phase. Whereas C & D vaccinate 90% of the 65+ and 90% of the most transmissible one. So as we consider only scenarios where we vaccinate 50% of the population, the benefit of cyclic vaccination are the one of not prioritizing a low mortality, low transmission group with these 50% of doses. Indeed, that the point the paper make, but I believe the context is which this is valid is quite restricted. It's less about the cyclicity but more that if you have doses for half the population, vaccinated 90% of the high risk & 90% of the high transmissibility is more efficient than vaccinating 90% of the high risk and 90% of the low risk & low transmissibility. The paper shows that "going down in age" for prioritisation is highly suboptimal (because age-ifr of verity et al is highly skewed towards elder). It is an interesting conclusion.

(
Tangentially, I **believe** the authors forgot to push the new code to github, hence it's hard for me to understand the details & reproduce the results. If this is true, I'll let the editor decide based on how much a concern this is with RSoS policies. I am especially interested in how reduction of immunity for vaccinated is implemented (ODE rate or fixed duration).
)

A concern I have (and would have checked in the code or if the behaviour of the vaccinated compartment was given) is:

- duration of immunity of one year (for most results*)
 - scenarios do a strategy for three months, then again. Repeated with a frequency that is less or equal than one year.
 - you do not keep track of who is vaccinated.
- > In this case, doesn't the vaccination strategy "switching" has an even exaggerated efficacy because it vaccinates different people each cycle (in it's second phase in figure S1) ? Whereas the strategies focused on one group revaccinate the same people over and over (some of which, but not all, having lost immunity, while the rest of the doses is wasted). If this is the case, I'd vouch for a memory of who is vaccinated (this is in line with what happens in most country). This might affects line 25. of page 28. Is this the case ?

* Figure 1 caption says that it shows results for 3 immunity duration, 3 transmission level, 3 vaccine efficacy and 3 vaccination frequencies and 3 assessment duration, and two strategies. I believe the 3 immunity duration are shown in the CI (which I find strange) but I do think it

would warrant its own degree of freedom in the graph (or as an additional graph, as fig 2. and 3. are very similar). Especially since authors do not keep track of who is vaccinated.

A minor nitpick is that when rotating the label of the x-axis, they are shifted to the left. This affects most figures in this paper.

Decision letter (RSOS-210292.R1)

Dear Dr Stenseth

On behalf of the Editors, we are pleased to inform you that your Manuscript RSOS-210292.R1 "Switching vaccination among target groups to achieve improved long-lasting benefits" has been accepted for publication in Royal Society Open Science subject to minor revision in accordance with the referees' reports. Please find the referees' comments along with any feedback from the Editors below my signature.

Please submit your revised manuscript and required files (see below) no later than 7 days from today's (ie 04-Jun-2021) date. Note: the ScholarOne system will 'lock' if submission of the revision is attempted 7 or more days after the deadline. If you do not think you will be able to meet this deadline please contact the editorial office immediately.

on behalf of Professor Enrico Bertuzzo (Associate Editor) and Pete Smith (Subject Editor)
openscience@royalsociety.org

Associate Editor Comments to Author (Professor Enrico Bertuzzo):

Comments to the Author:

The manuscript has now been reviewed by the two previous referees. Given their feedback, I believe that another quick round of revision would benefit the paper.

Regarding the criticism of the first reviewer, according to the journal policies it is ok to produce results that depend on specific (yet plausible) assumptions, as long as the methods are sound. This is the case of the manuscript, but I encourage the authors to expand the discussion on how the specific assumptions made could affect the results. Moreover, there are novel evidence that vaccine efficacy against infection could be as high as the one against disease. I suggest the authors to collect and cite the latest evidence that support their assumption.

Please make sure to update the code of github and accomodate the suggestions of the second reviewer on the interpretation of the results.

I will personally evaluate the revision.

Reviewer comments to Author:

Reviewer: 1

Comments to the Author(s)

The authors did answer on the technical aspects of my criticism, but they did not justify their assumptions, which I still find not realistic and which entail that the conclusions might be the opposite of the conclusions that one would draw if proper assumptions were made.

"we do not explicitly distinguish the efficacy against infectiousness and progression to severe cases and death": this does not answer my criticism on the fact that it seems more and more likely that most vaccines have a very strong protection against severe symptoms and death, while a rather low protection against being infectious. This is one key aspect of the analysis of this paper and cannot be neglected. I think that one cannot just assume that these two numbers coincide and I would rather expect a study on how several ratios impact the results obtained in this paper.

"To provide a general model and findings, we assume an all-or-nothing vaccine which provides perfect protection to a fraction of the vaccinated individuals.": I do not understand how making a (strong and most likely wrong) assumption would make the results more general. To me this is an arbitrary choice which limits the validity of your results to the case in which that assumption is correct. Since all evidence is telling that this assumption is likely to be wrong, I have to conclude that the obtained results do not apply to the COVID-19 pandemic.

"Additionally, to what extent vaccines could reduce the burden of mortality is also dependent on the vaccines supply. Even though prioritizing the at-risk groups is broadly adopted across countries, vaccine supply, in reality, varies vastly across countries. As expected, countries with sufficient doses to cover at least the at-risk groups would lead to a low level of mortality. In contrast, it is less likely to lower the burden of mortality in countries with very limited doses. Acknowledging this uneven distribution of vaccines, we keep a general structure of our model framework without explicitly distinguishing the efficacy against infectiousness and progression to severe cases and death.": I do not see any connection between this reasoning, which I share, and my criticism. If the vaccine supply is sufficient, then vaccinating people at risk is reducing mortality. If the vaccine supply is insufficient then the mortality will be higher. How does this relate to the findings of the manuscript?

Reviewer: 2

Comments to the Author(s)

I want to thank the authors for taking the time to account our reviews. I believe my points have been addressed, the (most) major one was that in the previous scenarios' definition, the total number of vaccines given was different between scenario (waste of doses). That's fixed now. Thank you.

Thanks for adding figure S1, which help a lot understanding the scenarios. Thanks also for adding figure S2. I believe Figure S1 & S2 are really important in judging why the scenario's impacts are so different. In S2, the two waves pattern exhibited due the initial conditions, with a really big first wave make the timing of the vaccination of high transmission very important, and drives the obtained results, short and long term. In Figure S1, Scenario A and B are suboptimal because they waste doses in on individuals with low transmission and fatality rate (40 to 65, as fatality comes from Verity et al) in their second phase. Whereas C & D vaccinate 90% of the 65+ and 90% of the most transmissible one. So as we consider only scenarios where we vaccinate 50% of the population, the benefit of cyclic vaccination are the one of not prioritizing a low mortality, low transmission group with these 50% of doses. Indeed, that the point the paper make, but I believe the context is which this is valid is quite restricted. It's less about the cyclicity but more that if you have doses for half the population, vaccinated 90% of the high risk & 90% of the high transmissibility is more efficient than vaccinating 90% of the high risk and 90% of the low risk & low transmissibility. The paper shows that "going down in age" for prioritisation is highly suboptimal (because age-ifr of verity et al is highly skewed towards elder). It is an interesting conclusion.

(
Tangentially, I ****believe**** the authors forgot to push the new code to github, hence it's hard for me to understand the details & reproduce the results. If this is true, I'll let the editor decide based on how much a concern this is with RSoS policies. I am especially interested in how reduction of immunity for vaccinated is implemented (ODE rate or fixed duration).
)

A concern I have (and would have checked in the code or if the behaviour of the vaccinated compartment was given) is:

- duration of immunity of one year (for most results*)
- scenarios do a strategy for three months, then again. Repeated with a frequency that is less or equal than one year.
- you do not keep track of who is vaccinated.

 In this case, doesn't the vaccination strategy "switching" has an even exaggerated efficacy because it vaccinates different people each cycle (in it's second phase in figure S1) ? Whereas the strategies focused on one group revaccinate the same people over and over (some of which, but not all, having lost immunity, while the rest of the doses is wasted). If this is the case, I'd vouch for a memory of who is vaccinated (this is in line with what happens in most country). This might affects line 25. of page 28. Is this the case ?

* Figure 1 caption says that it shows results for 3 immunity duration, 3 transmission level, 3 vaccine efficacy and 3 vaccination frequencies and 3 assessment duration, and two strategies. I believe the 3 immunity duration are shown in the CI (which I find strange) but I do think it would warrant it's own degree of freedom in the graph (or as an additional graph, as fig 2. and 3. are very similar). Especially since authors do not keep track of who is vaccinated.

A minor nitpick is that when rotating the label of the x-axis, they are shifted to the left. This affects most figures in this paper.

===PREPARING YOUR MANUSCRIPT===

===PREPARING YOUR REVISION IN SCHOLARONE===

-- If you have uploaded ESM files, please ensure you follow the guidance at <https://royalsociety.org/journals/authors/author-guidelines/#supplementary-material> to include a suitable title and informative caption. An example of appropriate titling and captioning may be found at https://figshare.com/articles/Table_S2_from_Is_there_a_trade-off_between_peak_performance_and_performance_breadth_across_temperatures_for_aerobic_scops_in_teleost_fishes_/3843624.

Author's Response to Decision Letter for (RSOS-210292.R1)

See Appendix B.

Decision letter (RSOS-210292.R2)

Dear Dr Stenseth,

I am pleased to inform you that your manuscript entitled "Switching vaccination among target groups to achieve improved long-lasting benefits" is now accepted for publication in Royal Society Open Science.

COVID-19 rapid publication process:

We are taking steps to expedite the publication of research relevant to the pandemic. If you wish, you can opt to have your paper published as soon as it is ready, rather than waiting for it to be published the scheduled Wednesday.

This means your paper will not be included in the weekly media round-up which the Society sends to journalists ahead of publication. However, it will still appear in the COVID-19 Publishing Collection which journalists will be directed to each week (<https://royalsocietypublishing.org/topic/special-collections/novel-coronavirus-outbreak>).

If you wish to have your paper considered for immediate publication, or to discuss further, please notify openscience_proofs@royalsociety.org and press@royalsociety.org when you respond to this email.

on behalf of Professor Enrico Bertuzzo (Associate Editor) and Pete Smith (Subject Editor)
openscience@royalsociety.org

Appendix A

Associate Editor Comments to Author (Professor Enrico Bertuzzo):

The manuscript has now been reviewed by two experts on the field. Both found that paper clearly written and sound. However, they both raised concerns about some hypotheses and asked to clarify some others. Both encourage a major revision to improve, or else justify, the analysis and I share their view.

Reviewer comments to Author:

Reviewer: 1

Comments to the Author(s)

This paper compares some vaccination strategies with the intent of reducing the mortality of COVID-19.

The approach seems reasonable and the paper is clearly written and well structured.

[General response] We appreciate the reviewer's insightful comments and suggestions, which have been very helpful in improving the manuscript. We have revised the manuscript carefully in light of the suggestions provided. Below, we provide point-by-point responses to the concerns of the reviewer.

My main concerns regard the assumptions made by the authors, since I believe the results strongly depend on them and I am not fully convinced that this setup is the most significant one:

- why is the low efficacy vaccine only 20% effective? It seems that vaccines having less than 60% efficacy will not be approved, at least in Europe. Moreover, most approved vaccines have an efficacy to prevent severe disease which is 100% or very close to that.

[Response] We consider an all-or-nothing vaccine which provides perfect protection to a fraction, defined by vaccine efficacy, of the vaccinated individuals. Indeed, current COVID-19 vaccines have a high efficacy against progression to severe cases and death. However, efficacy against infectiousness is still inclusive. To keep our models and findings as general as possible, we do not explicitly distinguish the efficacy against infectiousness and progression to severe cases and death. Nevertheless, we assume a 20% efficacy for the scenario of low vaccine efficacy in general. The flu vaccine efficacy can be as low as 20% in years of mismatch.

We have now clarified our assumptions of efficacy in the Methods and Figure S1 in the revised manuscript.

- I think that for this study it is fundamental to distinguish how much a vaccine protects against the severe disease and how much a vaccine protects against becoming infectious. Since it seems that the two numbers are different, this distinction cannot be avoided. Moreover, many currently available vaccines seem to have 100% efficacy (or close to that) against severe disease, while lower (even much lower, around 60-70% for some) efficacy against light disease, which might entail that the infected person is infectious. I think it is still a subject of debate to what extent currently available vaccines protect also against transmitting the infection. If this information were included in the proposed analysis (by taking some reasonable assumptions, given the lack of results), I am confident that the standard strategy of aiming at direct protection will turn out to be the best.

[Response] We agree with the reviewer that vaccine efficacy against infectiousness and progression to severe cases and death could be different. Related to your comment and our response above, we do not distinguish them as to what extent vaccine could reduce the infectiousness and progression to severe cases and death is still unclear. To provide a

general model and findings, we assume an all-or-nothing vaccine which provides perfect protection to a fraction of the vaccinated individuals. Accordingly, we do not explicitly model the number of deaths; rather, we estimate the mortality by multiplying the modelled infected fraction and the infection-fatality-ratio (IFR). Such that vaccine prevents the infections and thus reduces the mortality. A similar framework and analysis are also taken by Matrajt et al. (1) – which now is cited in our manuscript. Indeed, current COVID-19 vaccines have a high efficacy to prevent severe disease, suggesting that we may probably overestimate the mortality.

Additionally, to what extent vaccines could reduce the burden of mortality is also dependent on the vaccines supply. Even though prioritizing the at-risk groups is broadly adopted across countries, vaccine supply, in reality, varies vastly across countries. As expected, countries with sufficient doses to cover at least the at-risk groups would lead to a low level of mortality. In contrast, it is less likely to lower the burden of mortality in countries with very limited doses. Acknowledging this uneven distribution of vaccines, we keep a general structure of our model framework without explicitly distinguishing the efficacy against infectiousness and progression to severe cases and death.

Reference:

1. Laura Matrajt, Julia Eaton, Tiffany Leung, Elizabeth R. Brown. Vaccine optimization for COVID-19: Who to vaccinate first? *Sci. Adv.* 2021; 7: eabf1374

- I am not sure why the authors assume that only 50% of the population will be vaccinated, since most countries are aiming at vaccinating 100% of the population. Also, it is not fully clear to me what the different vaccination policies are. Maybe you can include an equation or an algorithm to explain that. Does one-off mean that you vaccinate 50% of the population and then stop vaccinating forever? For the yearly campaign does that mean that in 3 months you vaccinate 50% of the population, then wait for 9 months and start vaccinating again? In that case, do you assume to know who is still protected by the vaccine and who is not? It seems hard to make this distinction with the model you use, since some (few) people lose protection almost immediately with model (1)-(4). The same question is even stronger for the 6-month approach, since in that case you should have a moment in which 100% of the population is protected. Then how can the three strategies be distinguished? I think this point needs to be clearly explained since it is crucial to understand the results.

[Response] We acknowledge that vaccine coverage is a key epidemiological parameter and is highly relevant to our findings. It is true that some countries are aiming at a very high coverage. Since the coverage varies over time and locations, we assume a 50% coverage in order to give a general perspective. More importantly, we believe this is an ideal assumption with which we can implement our analysis and present our findings appropriately. As indicated in Matrajt et al., if the coverage is low (e.g. lower than ~50%), vaccinating the at-risk will always be the best option to minimize the mortality. This is suggestive of a coverage above ~50% for the issue of comparing different vaccine strategies. In line with our aim to keep a general framework, we use 50% in the analysis. This should be taken as a benchmark for the analysis with respect to vaccine strategies. To examine the impact of vaccine supply, we make additional simulations with lower (40%) and higher (90%) coverage of the population across countries. Results show a proven effectiveness of the cyclic strategy irrespective of total vaccine supplies (please refer to Figure S4).

We consider three vaccine frequencies. One-off vaccination means that we only vaccinate 50% of the population in the initial year and then stop vaccinating forever. For the annual vaccination, we assume it covers 50% of the population during the 3-month period in the first year and wait for 9 months and then start vaccinating again. Similarly, for the 6-month campaign, we assume the vaccination covers 50% of the population within three months in the first 6-month and wait for 3 months and then start vaccinating again. Indeed, it is expected

to have a moment when 100% of the population is protected as immunity durations could be longer than the duration of between consecutive campaigns. Nevertheless, we, for the annual and 6-month campaign, do not distinguish who is still protected by the vaccine and who is not in the next campaign. This means that people can get vaccinated in the following campaigns irrespective of their status of protection and even though 100% of the population could probably be protected. We believe this is closer to the realistic vaccination campaign.

We have now presented simulations with other vaccine supplies and clarified the definition of strategies and assumptions in the revised manuscript.

Minor comments:

- In (4) I suppose a "q" is missing, otherwise the total population is not preserved constant.
- planed -> planned

[Response] We have now corrected this typo.

- I disagree with the following conclusion: "In particular, if assuming a 1-year duration of immunity, we suggest that carrying out the cyclic vaccination every six month since the second year of initializing the vaccination campaign would be optimal to achieve approximately 21% (low-level transmission) and 9% (medium-to-high transmission) lower burden of mortality as compared to targeting the elderly only in the long-term."

Left aside the discussion on the validity of the assumptions, the only claim that can be made is that one of the studied strategies is better than the other studied ones. In order to claim optimality, one needs to formulate and solve an optimization problem, or to prove that (at least locally), there exists no strategy which performs better.

[Response] We did indeed not use optimization algorithm to prove the optimality of the proposed vaccination strategy. We have now tempered down this statement in the revised manuscript.

Reviewer: 2
Comments to the Author(s)

This work compared different age-based vaccination strategies for vaccines in different countries, for endemicity control. The paper is clearly written and to the point. The code is clear as well, and the findings are reproducible.

I do have major concerns that advise for at least major revision.

- The model assumes an equal absolute amount of contact for each age-class, which in this case influence the results greatly. The death process is not defined in the main text.
- Some assumptions, mainly regarding vaccine roll out function, initial conditions, are counterfactual. And the conclusion is dependent on these. However, the writing overstates the generalization and the practical importance of the findings. The scenarios seems to spend a different total amount of vaccine per day, making the comparison result obvious: the more we vaccinate the more it is effective (I'm not sure of that, so maybe the authors might correct me on this).

[General response] We appreciate the reviewer's insightful comments and suggestions, which have been very helpful in improving the manuscript. We have revised the manuscript carefully in light of the suggestions provided. Below, we provide point-by-point responses to the concerns of the reviewer.

Abstract:

Except if derived from the journal editorial policy, I think it would be interesting to have a hint into the results. The last sentence of the abstract may be re-written to be more comprehensible.

[Response] It is indeed necessary to highlight our findings appropriately. We convey two messages here. First, by comparing the strategies targeting on the high-risk groups only versus both the high-risk and high-contact groups simultaneously, we point that the latter leveraging the benefits from both the direct and indirect protection is crucial in reducing the burden of mortality. Moving beyond this, we propose that focusing on both groups in a consecutive manner, rather than simultaneously as adopted in the mixed vaccination, may be the key.

We have now re-written the abstract.

Introduction

L39: successive waves, as some of the 15 studied countries had more waves.

[Response] We agree with the reviewer and has now updated this in the revised manuscript.

L46: this can be argued to be DALYs too (e.g avoid two 95 y.o deaths may not be considered more important than avoiding a 40 y.o death). Doesn't matter in this context though.

[Response] DALYs is an important indicator of disease burden and thus the goal of vaccination. Nevertheless, we argue it to be mortality in this context.

L50: Matraj et al (somewhere on medRxiv last time I looked) also tackle this issue and are not cited in the paper (I'm not affiliated to them in any way).

The introduction is short and to the point, which is great. I do believe a bit more of literature could be cited on this kind of problem (vaccination of high transmission vs high death).

[Response] We thanks for the reviewer's suggestion. We have now enriched the introduction by including additional studies (1-2) on the discussion of to whom the vaccination should be prioritized. The Matrajt et al. study is now published in Sci Adv.

Reference:

1. Laura Matrajt, Julia Eaton, Tiffany Leung, Elizabeth R. Brown. Vaccine optimization for COVID-19: Who to vaccinate first? Sci. Adv. 2021; 7:eabf1374
2. Jack H. Bucknera, Gerardo Chowell, Michael R. Springborn. Dynamic prioritization of COVID-19 vaccines when social distancing is limited for essential workers. Proc. Natl. Acad. Sci. U.S.A. 2021; 118:e2025786118

Methods:

The two papers cited for age-specific contact are state of the art, however there has been some changes with the pandemic awareness, lowering the contact of the elder. It would be interesting (trough not necessary for me) to compare to these. One source is <https://cmmid.github.io/topics/covid19/synthetic-contact-matrices.html> (again, not affiliated to them in any way).

[Response] To examine the impact of the changing sociable behaviour during the pandemic, e.g. the lower contact of the elderly as suggested by the reviewer, we have now simulated the model by using the contact matrix in 2020. Considering that this changing human behaviour will not last long, we only evaluate the short-term effectiveness of different strategies. Overall, the reduction of mortality is lower across scenarios due to the less contacts during the pandemic. However, the findings are consistent, showing that integrating the direct and indirect protection would lead to a greater reduction of mortality.

We have now included this analysis and presented the results in Figure S5 in the revised manuscript.

L19: this wide prioritization for countries specified, e.g France focused more on the 75+ initially. This undermines a bit the i strategy, because authors consider

[Response] We define the elderly as 65+, accounting for four age groups i.e. people over 80, 75-79, 70-74 and 65-69. As suggest by your comment below, we now also consider the older individuals in the 45-64yrs if there is sufficient vaccine supply. By doing so, we base our analysis on every 20-year age group. Projecting our approaches to the finer, say every 10-year age group, will lead to the definition of the elderly group as 75+ which is consistent with the policy adopted in most countries initially. However, this 10-year range of ages may probably be enlarged so as to targeting more age groups in the sequential vaccination initiatives. In our study, we assume a consistent age interval over time, and we believe this 20-year age interval is ideal for the discussion of vaccination strategies. Either the narrower or broader age interval may undermine the findings with respect to the strategy issue.

L25: this sentence was not immediately clear to me. In fact, I think this whole paragraph should be made clear. Assumptions on vaccine efficacy and duration of protection may be stated here. You should also state here than you consider an all or nothing vaccine (by opposition of a leaky vaccine). This is not necessarily the consensus for COVID vaccines.

[Response] We consider an all-or-nothing vaccine which provides perfect protection to a fraction, defined by vaccine efficacy, of the vaccinated individuals. Additionally, we assumed different efficacies, i.e. the low, medium and high efficacy that protecting 20, 50, and 80% percent of the people who received vaccine. We assume the same average protected period ($1/w$) for the recovered and vaccinated individuals, meaning that their immunity may loss at the rate of w .

We have now clarified our assumption of the all-or-nothing vaccine, the efficacy and duration of protection in the revised manuscript.

The epidemic model is sound and does not consider useless features for the research questions, which is really good. The R_0 seems plausible for endemicity.

I may be wrong, but it seems that authors normalize the contact matrix, and from L46 it seems that beta isn't age-dependent. Hence, an assumption is made that every age-class has the same absolute number of contact (and with whom these contacts are is embedded in matrix C). This assumption is not realistic and some classes have order of magnitude the amount of contact (elderly being notoriously and unfortunately very isolated in these first-world countries, it is visible in these POLYmod matrices).

[Response] We normalize the age-specific contact matrix by the average contacts over a population. Such normalization means that $\beta \cdot C$ ends up with the weighted polymod matrices that have comparable, though highly age-dependent, force-of-infection on each population as a whole across the wide range of countries considered.

From the code, I gather that the model is initialized with .1% infected and zero recovered. This is quite unrealistic and should be at least stated in the main text.

[Response] We were not trying to fit the initial stage of the transmission in each country, while, for simplicity and better comparison among countries, we assume a 0.1% infected for all the countries. By doing so, we contextualize the model for each country by using the country-specific demography and social mixing patterns. This helps to present how the findings vary with the demography and mixing. We agree that the initial conditions (e.g. the proportion of susceptible and infected individuals in the population) could be higher than we assumed and differ across countries.

We have now clarified our assumptions in the revised manuscript.

Moreover, the death process and its dependence on age should be described.

[Response] We agree with the reviewer to clarify the death process. We did not explicitly model the death process; rather, we estimate the deaths by multiplying the modelled infections and the age-specific infection-fatality-ratio (IFR) (1). Such that both the infected fraction and IFR and thus the mortality is dependent on the age. We have now clarified this in the revised manuscript.

Reference:

1. R. Verity, L. C. Okell, I. Dorigatti, P. Winskill, C. Whittaker, N. Imai, et al. Estimates of the severity of coronavirus disease 2019: a model-based analysis. *Lancet Infect. Dis.* 20, 669–677 (2020). doi: 10.1016/S1473-3099(20)30243-7

Results

Depends on the assumptions: when there is no more susceptible because everyone in the age-class is vaccinated, vaccines are wasted in this model and this is not likely to happen in reality. Countries keep a basic accounting of attribution.

The cyclically switching should be done reactively to the proportion of vaccinated in each category (i.e. to keep the spirit of the paper, after the most lethal age-class is 100% (or 75% or 50%, that could be scenarios) vaccinated, move to the next age-class.

[Response] Indeed, vaccinating only the elderly or the most sociable ones may lead to the waste of the vaccine, in particular in the case of high coverage of vaccination. Given this, we have now refined the definition of the vaccination strategies. In line with the design of our study, strategy (i-a) and (i-b) focus on single groups, while strategy (ii-a) and (ii-b) focus on two groups. Strategy (i-a) target the high-risk older age groups (not just the elderly), meaning that we allocate the vaccination by the age ladder. That is, we initiate the vaccination by targeting to the elderly (65+, this is the group we defined previously), and then allocate the rest doses, if any, to the next high-risk age group (45-64yrs), and then 25-44. Similarly, Strategy (i-b) target the high-sociable groups (not just the most socially mixing ones), meaning that we allocate the vaccination by the ladder of contacts. Put in another way, we initiate the vaccination by targeting to the most socially mixing ones (this is the group we defined previously), and then allocate the rest doses, if any, to the next high-sociable group. Allocating the rest of vaccine doses also applies to the strategy (ii-a) and (ii-b).

We have now updated the definition of the strategies and the analyses in the revised manuscript.

The introduction teases a comparison between cyclical switching and mixed attribution, I have not seen these results.

[Response] We aim to convey two inter-linked messages in our study. First, we point that leveraging the benefits from both the direct and indirect protection is crucial in reducing the burden of mortality. To do this, we compare direct vaccination of the high-risk older age groups (i-a in main text) and the mixed vaccination of both the high-risk and core-sociable groups simultaneously (i.e. the mixed vaccination, ii-a in main text) (as shown in Figure 1). Moving beyond the benefits from mixed vaccination, we further consider alternative approach to leverage both the direct and indirect protection. That is, how the benefits from vaccinating both high-risk and high-contact individuals could be increased (i.e. through vaccinating them simultaneously or say cyclically). To this end, we examine the incremental reduction by using the cyclic strategy (ii-b in main text) as compared to the vaccination prioritized to the elderly only (i-a in main text) and the core sociable only (i-b in main text) in the short and long term (as shown in Figure 2-3). By doing so, the two messages are: integrating the benefits from direct and indirect protection is of proven potential to maximizing the societal health impact. More importantly, over time switching the priority from high-risk older age groups to core-sociable groups responsible for heightened circulation and thus indirect risk may be key.

We have now improved the logic of our analysis by demonstrating the two messages in the Abstract and clarifying the design of the study in the Methods in the revised manuscript.

I might be wrong (in which case this should be addressed in the text) but the main result here is dependent on the total throughput of vaccine, which seems different (from the methods) for the different for the scenarios considered. The practical vaccination problem a country has is: given a amount, where to allocate it. Obviously if the amount was dependent, vaccinating across every category in // is optimal. However, as of today, if a dose is given to an 80 y.o, it won't be given to a 40y.o.

[Response] We agree with the reviewer that the issue of vaccination allocation should be investigated with a fixed amount of total doses across strategies. In the analysis, we assume that the total doses is sufficient to cover 50% of the population (please see other coverage (40%, 90%) in the SI). However, our previous definition of different strategies may lead to the waste of the vaccine doses, such that even though we assume the same amount of vaccine across strategies, not all of the vaccine was fully allocated. We have now updated the definition of strategies (please refer to our response to your comment above), ensuring that all of the doses have been fully allocated.

Aside from the comment, it would be interesting to see a timeline of vaccination and epidemic in time, to see how the system is reacting and transitioning, in one setup. Like a graph with death and infection as Y and time as X, with vaccine rollout and % vaccinated.

[Response] To better present the design of our study, we have now presented the group-specific coverage in each strategy (please refer to Figure S1) and the trajectories of infections and mortality (please refer to Figure S2).

Hence, I would advise to display clearly, for each scenario and each category, how many doses are spent. If it is not equal, to change the problem so that it is equal across scenarios.

[Response] Since we updated the definition of the strategies, the total doses is same across scenarios which covers 40%, 50% or 90% of the population (please refer to our response to your comment above).

Appendix B

To the editorial office:

Thanks for your decision letter of May 18. We are somewhat confused by your feedback. From the editorial office we are told that our submission is rejected. However, the handling editor, Professor Enrico Bertuzzo, says “[t]he manuscript has now been reviewed by the two previous referees. Given their feedback, I believe that another quick round of revision would benefit the paper”. He furthermore closes his feedback message by saying that “I will personally evaluate the revision”. To us this implies that he is expecting a revision of our submission. We are furthermore encouraged by the very positive feedback provided by Referee #2 and the fact that we can deal with the comments provided by Referee #1 (and the comments provided by the handling editor on this reviewers comments on Referee #1’s comments).

As the editorial office has not provided any arguments for deviating from the recommendation provided by the handling editor, we chose to interpret the feedback that we, in effect, are invited to submit a revision of our submission.

Here we detail how we have dealt with the feedback, both from the editor and the referees.

Comments provided by the handling editor Professor Enrico Bertuzzo

The manuscript has now been reviewed by the two previous referees. Given their feedback, I believe that another quick round of revision would benefit the paper.

Regarding the criticism of the first reviewer, according to the journal policies it is ok to produce results that depend on specific (yet plausible) assumptions, as long as the methods are sound. This is the case of the manuscript, but I encourage the authors to expand the discussion on how the specific assumptions made could affect the results. Moreover, there are novel evidence that vaccine efficacy against infection could be as high as the one against disease. I suggest the authors to collect and cite the latest evidence that support their assumption.

Please make sure to update the code of github and accommodate the suggestions of the second reviewer on the interpretation of the results.

I will personally evaluate the revision.

[General response] We appreciate the editor’s and reviewer’s insightful comments and suggestions, which have been very helpful in improving our manuscript. We have revised the manuscript carefully in light of the suggestions provided.

Specifically we have:

- Expanded the discussion on how the specific assumptions made and how they might affect our results and conclusions.
- Cited novel evidence for vaccine efficacy
- Updated the code on github

Comments provided by Reviewer 1:

The authors did answer on the technical aspects of my criticism, but they did not justify their assumptions, which I still find not realistic and which entail that the conclusions might be the opposite of the conclusions that one would draw if proper assumptions were made.

[General response] We appreciate the reviewer's comments and suggestions, which have been very helpful in improving the manuscript. We have, as detailed below, revised the manuscript carefully in light of the suggestions provided.

"we do not explicitly distinguish the efficacy against infectiousness and progression to severe cases and death": this does not answer my criticism on the fact that it seems more and more likely that most vaccines have a very strong protection against severe symptoms and death, while a rather low protection against being infectious. This is one key aspect of the analysis of this paper and cannot be neglected. I think that one cannot just assume that these two numbers coincide and I would rather expect a study on how several ratios impact the results obtained in this paper.

[Response] We agree with the reviewer that vaccine efficacy against infectiousness and progression to severe cases and deaths could be different. However, as mass vaccination campaigns commence worldwide, novel evidence have shown that vaccines could be highly effective against SARS-CoV-2 infections across diverse populations in the real-world setting. For example, studies in Israel – the country with the largest coverage of vaccination – have indicated an effectiveness of >90% for both symptomatic and asymptomatic infections after the second dose (1-2). Similarly, studies in the US and UK have also provided evidence of vaccine effectiveness against SARS-CoV-2 infections among working-age adults (3-4). With such evidence it seems clear that vaccine provide strong protection against severe symptoms/deaths and being infectious. Therefore, we do not explicitly distinguish the efficacy in the analysis.

We have now justified our assumptions of efficacy with the novel evidence in the revised manuscript.

References:

1. Dagan N, Barda N, Kepten E, et al. BNT162b2 mRNA Covid-19 Vaccine in a Nationwide Mass Vaccination Setting. *N Engl J Med* 2021; 384:1412-1423.
2. Haas EJ, Angulo FJ, McLaughlin JM, et al. Impact and effectiveness of mRNA BNT162b2 vaccine against SARS-CoV-2 infections and COVID-19 cases, hospitalisations, and deaths following a nationwide vaccination campaign in Israel: an observational study using national surveillance data. *Lancet* 2021; 397: 1819–29.
3. Thompson MG, Burgess JL, Naleway AL, et al. Interim estimates of vaccine effectiveness of BNT162b2 and mRNA-1273 COVID-19 vaccines in preventing SARS-CoV-2 infection among health care personnel, first responders, and other essential and frontline workers — eight U.S. locations, December 2020–March 2021. *MMWR Morb Mortal Wkly Rep* 2021;70:495–500.
4. Hall V FFPH, Foulkes S, Saei A, et al. Effectiveness of BNT162b2 mRNA vaccine against infection and COVID-19 vaccine coverage in healthcare workers in England, multicentre prospective cohort study (the SIREN Study). Available at SSRN: <https://ssrn.com/abstract=3790399> or <http://dx.doi.org/10.2139/ssrn.3790399>

"To provide a general model and findings, we assume an all-or-nothing vaccine which provides perfect protection to a fraction of the vaccinated individuals.": I do not understand how making a (strong and most likely wrong) assumption would make the results more general. To me this is an arbitrary choice which limits the validity of your results to the case in which that assumption is correct.

Since all evidence is telling that this assumption is likely to be wrong, I have to conclude that the obtained results do not apply to the COVID-19 pandemic.

[Response] There are indeed two different types of vaccine responses, namely *all-or-nothing* and *leaky*, and thus two modelling approaches. The all-or-nothing model assumes that the vaccine either renders its recipient complete immunity or no immunity at all; while the leaky model assumes that vaccinated individuals respond by acquiring partial immunity. The various nuances of vaccine-induced immunity is laid out in Halloran ME (1). It is clear that only the future will show whether the new mRNA and adenovectored vaccine technology is sterilizing or leaky. We have now extended the discussion on this assumption and cited the reference in the revised manuscript.

Reference:

1. M. E. Halloran, I. M. Longini, C. J. Struchiner, *Design and Analysis of Vaccine Studies* (Statistics for Biology and Health, Springer, 2010).

"Additionally, to what extent vaccines could reduce the burden of mortality is also dependent on the vaccines supply. Even though prioritizing the at-risk groups is broadly adopted across countries, vaccine supply, in reality, varies vastly across countries. As expected, countries with sufficient doses to cover at least the at-risk groups would lead to a low level of mortality. In contrast, it is less likely to lower the burden of mortality in countries with very limited doses. Acknowledging this uneven distribution of vaccines, we keep a general structure of our model framework without explicitly distinguishing the efficacy against infectiousness and progression to severe cases and death." I do not see any connection between this reasoning, which I share, and my criticism. If the vaccine supply is sufficient, then vaccinating people at risk is reducing mortality. If the vaccine supply is insufficient then the mortality will be higher. How does this relate to the findings of the manuscript?

[Response] We thanks for the reviewer's suggestion. Factors other than vaccine efficacy, such as the uneven distribution of vaccines, may also be the key in reducing the burden of mortality. To keep a general structure of our model framework, we do not explicitly consider these factors. More importantly, with the evidence from recent studies on efficacy we believe that it is sound to consider the efficacy against severe cases/deaths and infections as strong. We have now revised the manuscript by referring to the evidence from recent studies on efficacy against infections (please refer to our response to your Comment #1).

Comments provided by Reviewer 2:

I want to thank the authors for taking the time to account our reviews. I believe my points have been addressed, the (most) major one was that in the previous scenarios' definition, the total number of vaccines given was different between scenario (waste of doses). That's fixed now. Thank you.

[General response] We appreciate the reviewer's kind comments and suggestions, which have been very helpful in improving the manuscript. We have revised the manuscript carefully in light of the suggestions provided.

Thanks for adding figure S1, which help a lot understanding the scenarios. Thanks also for adding figure S2. I believe Figure S1 & S2 are really important in judging why the scenario's impacts are so different. In S2, the two waves pattern exhibited due the initial conditions, with a really big first wave make the timing of the vaccination of high transmission very important, and drives the obtained results, short and long term. In Figure S1, Scenario A and B are suboptimal because they waste doses in on individuals with low transmission and fatality rate (40 to 65, as fatality comes from Verity et al) in their second phase. Whereas C & D vaccinate 90% of the 65+ and 90% of the most transmissible one. So as we consider only scenarios where we vaccinate 50% of the population, the benefit of cyclic vaccination are the one of not prioritizing a low mortality, low transmission group with these 50% of doses. Indeed, that the point the paper make, but I believe the context is which this is valid is quite restricted. It's less about the cyclicity but more that if you have doses for half the population, vaccinated 90% of the high risk & 90% of the high transmissibility is more efficient than vaccinating 90% of the high risk and 90% of the low risk & low transmissibility. The paper shows that "going down in age" for prioritisation is highly suboptimal (because age-ifr of verity et al is highly skewed towards elder). It is an interesting conclusion.

[Response] One motivation for a cyclic regime is to keep high-risk and high-contact from visiting vaccine-providing centers at the same time. Therefore, as the reviewer considered, the benefit of cyclic vaccination is the one of not prioritizing a low mortality, low transmission group. By contrast, focusing only single, either the high-risk or high-contact, group is suboptimal. This is because the infection-fatality ratio is much higher in the elderly (as the reviewer suggested). Additionally, it may be due to the fact that we will never achieve the herd immunity and the gains from indirect protection if we do not consider vaccinating the high-transmitters. Such findings is indeed relatively specific to COVID-19 as we have carefully tailored our model to the COVID-19 context. We have now briefly discussed this in the revised manuscript.

Tangentially, I ****believe**** the authors forgot to push the new code to github, hence it's hard for me to understand the details & reproduce the results. If this is true, I'll let the editor decide based on how much a concern this is with RSoS policies. I am especially interested in how reduction of immunity for vaccinated is implemented (ODE rate or fixed duration).

[Response] We thanks for the reviewer's suggestion and have now uploaded the new code to github (1) for you reference. We assume that immunity lasts for an average of $1/\omega$ days, and thus we assume an exponential reduction of immunity at the rate of ω .

Reference:

1. <https://github.com/ruiyunli90/Corona-vaccine>

A concern I have (and would have checked in the code or if the behaviour of the vaccinated compartment was given) is:

- duration of immunity of one year (for most results*)
- scenarios do a strategy for three months, then again. Repeated with a frequency that is less or equal than one year.

- you do not keep track of who is vaccinated.

 In this case, doesn't the vaccination strategy "switching" has an even exaggerated efficacy because it vaccinates different people each cycle (in its second phase in figure S1)? Whereas the strategies focused on one group revaccinate the same people over and over (some of which, but not all, having lost immunity, while the rest of the doses is wasted). If this is the case, I'd vouch for a memory of who is vaccinated (this is in line with what happens in most countries). This might affect line 25. of page 28. Is this the case?

[Response] While we do not track the vaccination status of each individual, we model fraction of individuals vaccinated by translating the total doses to the coverage of each targeted group. That is, we do not distinguish who is still protected by the vaccine and who is not in the following campaigns. This means that people can get vaccinated in the following campaigns irrespective of their vaccine-card status. However, this does not mean that, in each campaign, we will vaccinate different people and thus have an exaggerated efficacy by using the switching strategy; while revaccinate the same people and thus have a lower efficacy by using the strategy focused on single group. For each campaign, we have fixed the total number of doses given across scenarios and thus there is no waste of vaccine (as the reviewer has recognized). Accordingly, strategies focusing only on the high-risk older age groups includes the elderly over 65 years old, and, depending on the total vaccine doses, move to other groups by moving down the age ladder. By doing so, we will not revaccinate the elderly over and over.

Figure 1 caption says that it shows results for 3 immunity duration, 3 transmission level, 3 vaccine efficacy and 3 vaccination frequencies and 3 assessment duration, and two strategies. I believe the 3 immunity duration are shown in the CI (which I find strange) but I do think it would warrant its own degree of freedom in the graph (or as an additional graph, as fig 2. And 3. are very similar). Especially since authors do not keep track of who is vaccinated.

[Response] In figure 1, we present median estimates and 95%CI of the reduction of mortality across immunity durations for scenarios with differing transmission level, vaccine efficacy and vaccination frequency. We have now clarified this in the revised Figure 1 caption. Accordingly, we present the incremental reduction of mortality from indirect protection under the assumed 1-year immunity duration in Table 1. To clarify the difference across immunity duration, we have now provided Table S1-S3 to show the incremental reduction of mortality from indirect protection, assuming a 6-month, 5-year and permanent immunity duration, respectively.

A minor nitpick is that when rotating the label of the x-axis, they are shifted to the left. This affects most figures in this paper.

[Response] We have now relabelled the x-axis in all relevant figures.